# Allogeneic Stem Cell Transplantation in Multiple Myeloma

**Christine Greil, Monika Engelhardt** , **Jürgen Finke and Ralph Wäsch** *

University Medical Center Freiburg, Department of Hematology, Oncology and Stem Cell Transplantation, Faculty of Medicine, University of Freiburg, 79106 Freiburg, Germany; christine.greil@uniklinik-freiburg.de (C.G.); monika.engelhardt@uniklinik-freiburg.de (M.E.); juergen.finke@uniklinik-freiburg.de (J.F.)
* Correspondence: ralph.waesch@uniklinik-freiburg.de

**Simple Summary:** Due to its graft-versus-myeloma effect, allogeneic hematopoietic stem cell transplantation (allo-SCT) can enable long-term survival or even cure in carefully selected patients with multiple myeloma (MM), but remains controversial due to its relevant treatment-related toxicity. Current data suggest that allo-SCT should be considered in young MM-patients without relevant comorbidities in case of a high-risk constellation according to cytogenetics or stage, primarily as part of a tandem approach with autologous-SCT followed by allo-SCT and early in the course of the disease. Prospective studies are warranted, due to a suspected synergism especially those including new immunotherapeutic approaches for induction, conditioning and maintenance therapy.

**Abstract:** The development of new inhibitory and immunological agents and combination therapies significantly improved response rates and survival of patients diagnosed with multiple myeloma (MM) in the last decade, but the disease is still considered to be incurable by current standards and the prognosis is dismal especially in high-risk groups and in relapsed and/or refractory patients. Allogeneic hematopoietic stem cell transplantation (allo-SCT) may enable long-term survival and even cure for individual patients via an immune-mediated graft-versus-myeloma (GvM) effect, but remains controversial due to relevant transplant-related risks, particularly immunosuppression and graft-versus-host disease, and a substantial non-relapse mortality. The decreased risk of disease progression may outweigh this treatment-related toxicity for young, fit patients in high-risk constellations with otherwise often poor long-term prognosis. Here, allo-SCT should be considered within clinical trials in first-line as part of a tandem approach to separate myeloablation achieved by high-dose chemotherapy with autologous SCT, and following allo-SCT with a reduced-intensity conditioning to minimize treatment-related organ toxicities but allow GvM effect. Our review aims to better define the role of allo-SCT in myeloma treatment particularly in the context of new immunomodulatory approaches.

**Keywords:** multiple myeloma; allogeneic stem cell transplantation; immunotherapy; graft-versus-host disease

## 1. Introduction

Multiple myeloma (MM) is a heterogeneous disease and the second most common hematological malignancy [1]. It is characterized by the clonal expansion of malignant plasma cells in the bone marrow and associated with an overproduction of complete or incomplete monoclonal immunoglobulins [2]. The disease typically evolves from a monoclonal gammopathy of unknown significance (MGUS) to a smoldering MM (SMM) before becoming symptomatic due to displacement of normal hematopoiesis, destroyed bone structure, high monoclonal immunoglobulin levels and secondary immunodeficiency [3].

Based on the serum albumin and ß2-microglobulin levels and distinct cytogenetic aberrations [4], patients are stratified into different prognostically relevant risk groups according to the revised International Staging System (R-ISS) [5]. A risk-adapted treatment should be initiated with the occurrence of CRAB or SLiM criteria (hypercalcemia, renal impairment, anemia, bone lesions and/or more than 60% bone marrow plasma cells, a ratio of involved to uninvolved serum free light chains ≥100, more than one focal lesion

in magnetic resonance imaging) [6] and can induce substantial responses and improve long-term survival [7], especially in young and fit patients. According to the European Society for Blood and Marrow Transplantation (EBMT) guidelines, high-dose chemotherapy followed by autologous hematopoietic stem cell transplantation (auto-SCT) is the standard of care for these transplant-eligible patients with newly diagnosed MM [8]. Over the last decades, new effective therapeutic agents were developed, especially for elderly patients with relevant comorbidities ineligible for auto-SCT and those with relapsed and/or refractory multiple myeloma (RRMM) [9–11], including immunomodulatory drugs (IMID), proteasome inhibitors (PI), monoclonal antibodies, inhibitors of histone deacetylases, bispecific antibodies, chimeric antigen receptor T (CAR-T) cells and others [7,12–15]. Due to this remarkable increase of treatment options, and thus an often much deeper remission after optimized first-line therapy and the availability of effective salvage therapies, survival of MM patients has substantially improved over the last years [16–18]. However, with a median overall survival (OS) of 5 years, the outcome can be more dismal especially in high-risk (HR) constellations and leaves room for improvements [19,20]. By means of an immune-mediated graft-versus-myeloma (GvM) effect [21], allogeneic hematopoietic stem cell transplantation (allo-SCT) may enable prolonged progression free survival (PFS) and even cure. It is considered a clinical option for selected HR patients with RRMM, but also as consolidation after first-line induction under specific conditions [5,6]. Nevertheless, allo-SCT is controversially discussed because of its potential toxicity, the risk of graft-versus-host disease (GvHD) and a considerable treatment-related mortality (TRM). Interestingly, the number of transplantations increased in the last decades [16], but dropped again in the last years consistent with the development of numerous new therapeutic approaches. Due to those encouraging new treatment options and its high TRM some experts would not consider allo-SCT in MM anymore. However, it may still have a place especially in combination with those new immunotherapeutic approaches. Clear treatment guidelines are lacking, as there are only few prospective trials and retrospective analyses were often conducted in heterogeneous patient cohorts with discrepancies in conditioning therapies, in GvHD prophylaxes and in follow-up treatment, including donor lymphocyte infusions (DLI) and immunosuppressive interventions. The application of new substances in the post-transplant setting as consolidation or maintenance therapy or in case of relapse is of special interest, as synergistic immunomodulatory effects are expected to be induced. Allo-SCT may also be discussed to sustain response, i.e., after CAR-T cell treatment. Clinical trials investigating these questions are highly warranted. In this review, we discuss the role of allo-SCT in MM on the basis of available data, also in the context of these new immunotherapeutic strategies.

## 2. Allogeneic Transplantation in Newly Diagnosed and Relapsed and/or Refractory Myeloma

Allo-SCT with high-dose myeloablative conditioning (MAC) regimens has been performed for MM since the 1980s, mainly in patients younger than 50 years with RRMM, but was initially associated with a high therapy-related toxicity and TRM of 40 to 60% [22]. Survival rates significantly improved from 40 to 60% at two years already in the 1990s because of a reduced TRM due to optimization of supportive therapy, fewer infectious complications, earlier allo-SCT and less prior chemotherapy. Long-term survival was achieved in 10 to 25% of the patients and the plateau in survival curves indicated the curative potential of this therapeutic approach in selected patients [22]. In the following years, myeloablation achieved through high-dose chemotherapy and auto-SCT with maximal reduction of MM-cells was separated from allo-SCT with less myelosuppressive but highly immunosuppressive reduced-intensity conditioning (RIC) regimens to prevent treatment-related organ toxicities but allow a sufficient engraftment and GvM effect [23]. Several prospective trials demonstrated improved OS and PFS after this auto/allo-SCT approach with RIC in the first-line setting as compared to the control arm, mostly tandem auto-SCT, and randomization according to the availability of a human leukocyte antigen

(HLA)-identical donor [24–26] (Table 1). In two studies, prolonged PFS was shown at least in patients with HR cytogenetics [4,27–29] and no study demonstrated inferiority of the auto/allo-SCT arm [30–34], suggesting that HR constellations may be overcome by the allo-SCT.

All studies proved long-term survival in a subset of patients, with OS- and PFS-rates of 44% and 19% at ten years, respectively, in a pooled analysis of four prospective trials [35]. In this analysis, long-term OS was significantly better in the allo-SCT-arm [35]. However, some trials showing superior PFS but similar OS indicate that the increased TRM may probably counteract the benefit of a reduced relapse rate by allo-SCT [29]. TRM-rates remained as substantial with 20% at 10 years [35], but were not worse as compared to the auto-SCT control arm in more than half of the studies [24–26,28,29,32]. The leading cause of death was organ failure or an infectious complication and in only 6% GvHD [17,29].

Randomized trials comparing allo- with auto-SCT in salvage situations are missing. A prospective trial investigating the feasibility of allo-SCT in patients relapsing after auto-SCT showed an OS-rate of 74% at two years and a 1-year TRM of 26% [36]. Due to the heterogeneity of the analyzed cohorts, the available retrospective studies led to divergent results (Table 2): Most analyses suggest an improvement of PFS or lower relapse rate after allo-SCT, but a comparable or even inferior OS-rate due to relevant TRM [37–40]. In two earlier analyses survival was worse after allo-SCT as compared to a second auto-SCT [41,42]. In a study distinguishing different risk groups, similar results were observed for intermediate-risk patients defined by prognostic factors like their response to prior therapies and the response duration after their first-line therapy [43]. In contrast, a recent study revealed an improved OS despite a higher TRM-rate after allo-SCT [44].

Retrospective analyses comparing newly diagnosed vs. RRMM showed an improved survival when allo-SCT was performed in an earlier course of the disease, upfront or as part of an auto/allo-approach, and not as a salvage and/or very late-line therapy [16,45], and that the auto/allo- may be better than an upfront allo-SCT-alone approach [16]. Compatible with this, survival was dismal in patients relapsing after prior auto-SCT [36].

The desired survival benefit after allo-SCT has to be balanced against possible long-term or late onset side effects due to immunosuppression and GvHD influencing patients' quality of life. An objective assessment of these therapy-associated restrictions and long-term side effects is rarely implemented in clinical trials and, especially in retrospective analyses, quality of life is difficult to quantify. With the help of our revised Myeloma Comorbidity Index (R-MCI) we could show that quality of life may not necessarily be impaired after allo-SCT, probably because a reduction of illness-induced limitations may outweigh therapy-associated impairment [17,46]. However, long-term side effects of allo-SCT widely vary between individual patients and have to be seen as a dynamic process with changing burden of symptoms [47]. Thus, depending on the timepoint of symptom assessment, the rate of chronic GvHD of any grade ranges from 22% to 67% in different trials [17,24,37,44,48,49].

Due to the intensity of the treatment and expected side effects allo-SCT in general is only discussed in young, fit patients. However, the therapy decision is rarely taken on the basis of a standardized assessment of fitness and health condition but a subjective evaluation and careful consideration of risk factors by the attending physician. Patients with severe comorbidities are generally excluded from prospective clinical trials, and only patients under 65 to 70 years of age were included with a median age of 55 years [35]. Thus, there is a lack of concrete recommendations which patient may benefit most from allo-SCT. The use of comorbidity tools such as the transplantation-comorbidity index (HCT-CI) [50] to objectify the physicians' assessment and treatment decisions are highly recommended, also when allo-SCT is conducted outside of clinical trials.

## 3. Conditioning Therapy

Due to its substantial therapy-related toxicity, in earlier years, survival after MAC-was inferior as compared to RIC-regimens [18]. However, a recent pooled data analysis of 61 trials revealed no difference between MAC and RIC [51], probably due to the improved supportive therapies [52]. Again, there is a lack of randomized trials comparing different conditioning regimens. In retrospective analyses, the investigated protocols appear equivalent regarding survival and toxicity [48,53]. In clinical routine, the most frequently applied protocols consist of intermediate doses of anti-myeloma substances, mostly a combination of fludarabine and melphalan at a dose of 90–150 mg/m$^2$ and 140 mg/m$^2$, respectively, but data from three prospective first-line trials indicate that conditioning with total body irradiation can also be performed [24,29,33].

In most prospective studies randomization depended on the availability of an HLA-identical donor, thus, the impact of HLA-status on survival has not been examined. In a recent evaluation of registry data, the outcome of MM-patients receiving peripheral blood stem cells of HLA-matched vs. -mismatched donors and those receiving cord blood stem cells was similar [54]. However, in a multivariate analysis of a single-center study transplantation from a HLA-mismatched donor was a predictor of reduced survival after allo-SCT [55]. Of note, the number of haploidentical transplantations for the treatment of hematological malignancies has increased in the last years and it seems effective with tolerable toxicity, especially with post-transplantation GvHD-prophylaxis with cyclophosphamide. There are few data about haploidentical allo-SCT in MM, but small retrospective studies show that it is feasible with moderate TRM- and similar PFS-rates as compared to allo-SCT with HLA-matched donors [56–60].

**Table 1.** Overview of prospective trials on allo-SCT in MM.

| Source Paper | Therapy Line *Comparison* | # of pts. allo-SCT vs. Control | Conditioning | OS / PFS / TRM | allo-SCT vs. Control (Long-Term Data) | Prognostic Factors for Better Survival; *Further Results* |
|---|---|---|---|---|---|---|
| Costa et al., 2020 [35] | first-line pooled analysis of 4 trials *auto/allo- vs. (tandem) auto-SCT* | 899 vs. 439 | see single trials | 44 vs. 36% (10 ys) *<br>19 vs. 14% (10 ys) n.s.<br>20 vs. 8% (10 ys) *** | | *post-relapse survival 51 vs. 37% (5 ys) **** |
| Holstein et al., 2020 [61] | first-line (auto/allo-SCT) | 49 | fludarabine 150 mg/m$^2$, cyclophosphamide 1.5 g/m$^2$ | median 6.6 ys<br>median 3.6 ys<br>2% (6 mo) | | |
| Ahmad et al., 2016; Le Blanc et al., 2020 [26,62] | first-line *auto/allo- vs. auto-SCT (retrospective cohort)* | 92 vs. 81 | fludarabine 150 mg/m$^2$, cyclophosphamide 1.5 g/m$^2$ | 61 vs. 37% (10 ys) ***<br>41 vs. 21% (10 ys) ***<br>9 vs. 2% (10 ys) n.s. | | cGvHD; *no difference in post-relapse survival* |
| Krishnan et al., 2011; Giralt et al., 2020 [29,34] | first-line, SR/HR (β2-MG > 3 mg/L, del13q) *randomized: auto/allo-SCT vs. tandem auto-SCT* | 189/37 vs. 436/48 | TBI 2 Gy | SR: 44 vs. 43% n.s.;HR: 37 vs. 29% (10 ys) n.s<br>SR: 18 vs. 19% n.s.; HR: 21 vs. 4% (10 ys) *<br>SR: 20 vs. 11% ***; HR: 22 vs. 11% (10 ys) n.s. | | *post-relapse survival in SR better after allo-SCT *; no difference in HR* |
| Knop et al., 2019 [28] | first-line HR (del13q) *randomized: auto/allo- vs. tandem auto-SCT* | 126 vs. 73 | fludarabine 90 mg/m$^2$, melphalan 140 mg/m$^2$ | median 70 vs. 72 mo n.s.<br>median 35 vs. 22 mo **<br>14 vs. 4% (2 ys) ** | | |
| Bruno et al., 2007; Giaccone et al., 2011 and 2018 [24,63,64] | first-line *randomized: auto/allo-SCT vs. any treatment* | 58 vs. 46 | TBI 2 Gy | median 11.4 vs. 3.9 ys **<br>median 3.6 vs. 1.5 ys ***<br>10 vs. 2% (2 ys) n.s. | | *post-relapse survival median 7.5 vs. 2 ys *, difference most distinct in cohort with donor lymphocyte infusions* |
| Green et al., 2017 [65] | single-arm *first-line HR vs. RRMM (auto/allo-SCT with PI-maintenance)* | 24 vs. 7 | TBI 2 Gy with/without fludarabine 90 mg/m$^2$ | 61 vs. 29% (4 ys)<br>52 vs. 14% (4 ys)<br>8 vs. 14% (2 ys) | | |
| Björkstrand et al., 2011; Gahrton et al., 2013 [25,66] | first-line *randomized: auto/allo- vs. (tandem) auto-SCT* | 108 vs. 249 | TBI 2 Gy, fludarabine 90 mg/m$^2$ | 49 vs. 36% (8 ys) *<br>22 vs. 12% (8 ys) *<br>13 vs. 3% (3 ys) *** | | |
| Lokhorst et al., 2012 [33] | first-line *randomized: auto/allo-SCT vs. any treatment* | 122 vs. 138 | TBI 2 Gy | 55 vs. 55% (6 ys) n.s.<br>28 vs. 22% (6 ys) n.s.<br>16 vs. 3% (6 ys) ** | | |

**Table 1.** *Cont.*

| Garban et al., 2006; Moreau et al., 2008 [30,31] | first-line HR (β2-MG > 3 mg/L, del13q) *randomized: auto/allo- vs. tandem auto-SCT* | 65 vs. 219 | busulfan 4 mg/kg, fludarabine 125 mg/m$^2$ | median 34 vs. 48 mo [n.s.] | |
| --- | --- | --- | --- | --- | --- |
| | | | | median 19 vs. 22 mo [n.s.] | |
| | | | | 11 vs. 5% | |
| Rosinol et al., 2008 [32] | first-line *randomized: auto/allo- vs. tandem auto-SCT* | 25 vs. 85 | fludarabine 125 mg/m$^2$, melphalan 140 mg/m$^2$ | 62 vs. 60% (5 ys) [n.s.] | |
| | | | | 61 vs.35% (5 ys) [n.s.] | |
| | | | | 16 vs. 5% [n.s.] | |
| Kröger et al., 2002 [36] | RRMM | 21 | fludarabine 150 mg/m$^2$, melphalan 100–140 mg/m$^2$ | 74% (2 ys) | no relapse after prior auto-SCT |
| | | | | 53% (2 ys) | |
| | | | | 26% (12 mo) | |

Abbreviations: pts = patients; OS = overall survival; PFS = progression-free survival; TRM = treatment-related mortality; auto-/allo-SCT = autologous/allogeneic hematopoietic stem cell transplantation; RRMM = relapsed and/or refractory multiple myeloma; TBI = total body irradiation; Gy = gray; HR = high-risk; SR = standard-risk; mo = months; ys = years; n.s. = not significant; * $p < 0.05$; ** $p < 0.01$; *** $p < 0.001$; cGvHD = chronic graft-versus-host disease; PI = proteasome inhibitor.

**Table 2.** Overview of retrospective trials on allo-SCT in MM, published in the last 5 years.

| Source Paper | Therapy Line *Comparison* | # of pts. allo-SCT vs. Control | Conditioning | OS / PFS / TRM | allo-SCT vs. Control (Long-Term Data) | Prognostic Factors for Better Survival; *Further Results* |
| --- | --- | --- | --- | --- | --- | --- |
| Luoma et al., 2021 [52] | first-line (upfront, auto/allo-SCT) and RRMM | 205 | NMA-, MAC- and RIC-regimens with/without TBI | median 7.4 ys | | lower stage, cytogenetic SR, MAC, first-line, cGvHD, no aGvHD |
| | | | | median 1.8 ys | | |
| | | | | 8% (5 ys) | | |
| Jürgensen-Rauch et al., 2021 [67] | first-line (upfront, auto/allo-SCT) and RRMM | 37 | fludarabine 125 mg/m$^2$, cyclophosphamide 2 g/m$^2$ | 44% (10 ys) | | earlier therapy line, response prior to allo-SCT, GvHD |
| | | | | 44% (10 ys) | | |
| | | | | 9% (5 ys) | | |
| Gagelmann et al., 2021 [68] | first-line *auto/allo- vs. single/tandem auto-SCT* | 72 vs. 446/105 | RIC | 67 vs. 51/60% (5 ys) [n.s.] | | *for t(4;14) single auto-SCT worse, for del(17p) no difference* |
| | | | | 34 vs. 17/33% (5 ys) * | | |
| | | | | 10 vs. 1/4% (5 ys) | | |
| Shouval et al., 2020 [55] | RRMM | 100 | RIC-regimens | 18% (5 ys) | | normal albumin, low LDH, normal renal function, lower stage, matched donor |
| | | | | 17% (5 ys) | | |
| | | | | 36% (5 ys) | | |
| Park et al., 2020 [69] | RRMM | 24 | RIC | 44 % (2 ys) | | earlier therapy line |
| | | | | 29% (2 ys) | | |
| | | | | 38% (12 mo) | | |
| Eisfeld et al., 2020 [70] | first-line and RRMM | 90 | MAC- and RIC-regimens | 39% (5 ys) | | earlier therapy line; *prolonged immunoparesis as indicator for impaired survival* |
| | | | | 25% (5 ys) | | |
| | | | | 28% (5 ys) | | |
| Gran et al., 2020 [53] | first-line and RRMM *treosulfan-based vs. other RIC vs. MAC* | 508 vs. 2830 vs. 1177 | treosulfan-based vs. other RIC vs. MAC | 62 vs. 57 vs. 47% (5 ys) * | | *survival data for first-line patients, no difference in later therapy lines* |
| | | | | 32 vs. 33 vs. 32% (5 ys) [n.s.] | | |
| | | | | 10 vs. 17 vs. 19% (5 ys) [n.s.] | | |
| Chhabra et al., 2020 [71] | first-line and RRMM (relapsed after allo-SCT) | 137 (60) | NMA-, MAC- and RIC-regimens with/without TBI | 60% (5 ys) | | better post-relapse survival for SR, interval between allo-SCT and relapse >12mo, no aGvHD before relapse |
| | | | | 39% (5 ys) | | |
| | | | | 20% (5 ys) | | |
| Golos et al., 2020 [72] | first-line and RRMM | 60 | MAC- and RIC-regimens | median 23 mo | | cGvHD |
| | | | | median 9 mo | | |
| | | | | 57% | | |
| Hayden et al., 2020 [18] | first-line and RRMM *RIC vs. NMA vs. MAC vs. auto/allo-SCT* | 169 vs. 69 vs. 65 vs. 41 | NMA-, MAC- and RIC-regimens with/without TBI | 39 vs. 45 vs. 19 vs. 34% (5 ys) | | response prior to allo-SCT; *OS after MAC worse, esp. before 2002 ** * |
| | | | | 15 vs. 17 vs. 14 vs. 15% (5 ys) | | |
| | | | | 17 vs. 19 vs. 33 vs. 10% (5 ys) | | |
| Bryant et al., 2020 [73] | RRMM | 73 | busulfan 8 mg/kg, melphalan 140 mg/m$^2$, fludarabine 125 mg/m$^2$ | 50% (3 ys) | | lower stage, younger age, no GvHD, earlier therapy line |
| | | | | 30% (3 ys) | | |
| | | | | 22% (12 mo) | | |
| Ikeda et. al., 2019 [43] | RRMM *allo-SCT vs. 2. auto-SCT* | 192 vs. 334 | MAC- and RIC-regimens | OS all: 24 vs. 34% (5 ys) OS intermediate risk according adverse factors: 22 vs. 28% (5 ys) ** | | *adverse factors for OS in both groups: male, no response prior to SCT, short response after first-line, low performance status* |
| Greil et al., 2019 [17] | first-line and RRMM | 109 | RIC-regimens | 26% (10 ys) | | first-line, response prior to/after allo-SCT, cytogenetic SR; *quality of life not impaired* |
| | | | | 20% (10 ys) | | |
| | | | | 12% (10 ys) | | |

**Table 2.** *Cont.*

| | | | | | |
|---|---|---|---|---|---|
| López-Corral et al., 2019 [74] | first-line and RRMM | 126 | MAC- and RIC-regimens with/without TBI | 43% (5 ys) / 18% (5 ys) / 32% | relapse >6mo after allo-SCT, cGvHD; *similar responses to PI and IMID pre-and post-allo-SCT* |
| Fiorenza et al., 2019 [75] | RRMM | 74 | RIC-regimens | 29% (2 ys) / 46% (2 ys) / - | younger age, response prior to allo-SCT, interval between auto- and allo-SCT <12 mo |
| Rotta et. al., 2009; Maffini et al., 2019 [76,77] | first-line auto/allo-SCT | 244 | TBI 2 Gy, fludarabine 90 mg/m$^2$ | 41% (10 ys) / 19% (10 ys) / 14% (5 ys) | response prior to allo-SCT, SR, MRD-negativity by flow cytometry after allo-SCT |
| Maymani et al., 2019 [48] | first-line and RRMM *conditioning regimens* | 73 | busulfan/fludarabin vs. fludarabin/melphalan 100 vs. 140 mg/m$^2$ | 39 vs. 43 vs. 32% (3 ys) n.s. / 16 vs. 26. vs. 11% (3 ys) n.s. / 21 vs. 28 vs. 24% (3 ys) n.s. | cytogenetic SR, first-line |
| Kawamura et al., 2018 [78] | first-line and RRMM | 65 | MAC- and RIC-regimens with/without TBI | 47% (3 ys) / 10% (3 ys) / 23% (3 ys) | response prior to allo-SCT, younger age |
| Htut at al., 2018 [44] | first-line and RRMM *auto/allo- vs. tandem auto-SCT* | 264 vs. 558 | MAC- and RIC-regimens with/without TBI | 44 vs. 35% (6 ys) * / - / 6 vs. 1% (12 mo) ** | participation in clinical trial, male, novel agents at induction; *post-relapse survival 44 vs. 35% (6 ys) \** |
| Yin et al., 2018 [51] | pooled analysis of 61 trials first-line and RRMM | 8698 | NMA-, MAC- and RIC-regimens with/without TBI | 46% (5 ys) / 27% (5 ys) / 27% (5 ys) | first-line, response prior to allo-SCT; *auto/allo- and tandem auto-SCT in SR idem, survival of cytogenetic SR/HR and RIC/MAC idem* |
| Schneidawind et al., 2017 [79] | RRMM | 41 | NMA-, MAC- and RIC-regimens with/without TBI | 51% (3 ys) / 15% (3 ys) / 20% (3 ys) | *survival worse in case of allo-SCT after 2. auto-SCT, post-relapse survival better after IMID/PI* |
| Sobh et al., 2017 [54] | RRMM after 1–2 auto-SCT *matched vs. mismatched donor vs. cord blood stem cells* | 419 vs. 93 vs. 58 | RIC-regimens with/without TBI | 33 vs. 39 vs. 25% (5 ys) n.s. / 14 vs. 27 vs. 4% (5 ys) n.s. / 28 vs. 35 vs. 27% n.s. | |
| Montefusco et al., 2017 [80] | first-line and RRMM | 71 | MAC- and RIC-regimens with/without TBI | 60% (5 ys) / 39% (5 ys) / 12% (5 ys) | younger age, response prior to allo-SCT; *median post-relapse PFS with IMID/PI 7–14 mo* |
| Rasche et al., 2016 [81] | first-line and RRMM | 155 | RIC-regimens with/without TBI | median 53 mo / median 14 mo / 16% (d100) | first-line, response prior to allo-SCT, no extramedullary disease, no loss of donor chimerism; *survival of cytogenetic SR/HR idem* |
| Dhakal et al., 2016 [82] | first-line and RRMM | 77 | NMA-, MAC- and RIC-regimens with/without TBI | 64% (3 ys) / 47% (3 ys) / 13% (12 mo) | younger age, response prior to allo-SCT, no CMV-reactivation; *survival of cytogenetic SR/HR and MRD-neg/pos by flow cytometry idem* |
| Sobh et al., 2016 [16] | first-line and RRMM *before/after 2004 upfront vs. auto/allo-SCT vs. RRMM* | 1924 vs. 2004 vs. 3405 | NMA-, MAC- and RIC-regimens with/without TBI | early: 38 vs. 51 vs. 25%; late: 42 vs. 54 vs. 33% (5 ys) / early: 24 vs. 28 vs. 10%; late: 27 vs. 32 vs. 25% (5 ys) / early: 36 vs. 19 vs. 25%; late: 30 vs. 19 vs. 29% (3 ys) | |
| Franssen et al., 2016 [45] | first-line and RRMM *first-line (upfront, auto/allo-SCT) vs. RRMM* | 58 vs. 89 | NMA-, MAC- and RIC-regimens with/without TBI | median n.r. vs. 29 mo *** / median 30 vs. 8 mo *** / 16 vs. 19% (10 ys) n.s. | relapse >18 mo after auto-SCT, response prior to allo-SCT; *survival of cytogenetic SR/HR idem* |

Abbreviations: pts = patients; OS = overall survival; PFS = progression-free survival; TRM = treatment-related mortality; auto-/allo-SCT = autologous/allogeneic hematopoietic stem cell transplantation; RRMM = relapsed and/or refractory multiple myeloma; NMA = nonmyeloablative conditioning; MAC = myeloablative conditioning; RIC = reduced-intensity conditioning; TBI = total body irradiation; HR = high-risk; SR = standard-risk; mo = months; ys = years; d = day; n.s. = not significant; * $p < 0.05$; ** $p < 0.01$; *** $p < 0.001$; a/cGvHD = acute/chronic graft-versus-host disease; PI = proteasome inhibitor; IMID = immunomodulatory drug; MRD = minimal residual disease, n.r. = not reached.

## 4. Prognostic Factors

Due to the unproven survival advantage, allo-SCT is not considered as a standard of care in MM-patients. However, it should be discussed individually especially in younger patients without relevant comorbidities diagnosed with HR MM in the initial course of therapy, when the risk of progression may outweigh the transplant-related disadvantages [51], and allo-SCT may allow long-term survival with preserved quality of life [17].

Retrospective analyses revealed several prognostic factors that may be helpful for an individual risk-benefit assessment (Table 2).

As discussed above, the outcome of patients transplanted in the first-line setting or at least earlier in the course of their disease was significantly better than that of RRMM-patients after multiple therapy lines [43,44,54–58].

In various studies remission status at allo-SCT was also a relevant predictor for survival with significantly longer OS and/or PFS in patients responding to induction as compared to those with progressive disease at the time point of transplantation [17,18,36,43, 45,51,67,75,77,78,80–82]. The role of minimal residual disease (MRD)-status was analyzed in the post-transplant setting and is not conclusively clarified at this time: Achievement of MRD-negativity by flow cytometry after transplantation led to a survival benefit in a large retrospective analysis [76], whereas another trial could not prove a difference [82]. Similarly, a prolonged post-transplant immunoparesis was described as an indicator for dismal survival [70]. If allo-SCT is not conducted in terms of a tandem auto/allo-approach, the duration of response to prior therapy, especially to prior auto-SCT, plays a crucial role with a dismal prognosis in case of a less prolonged response [43,45,71].

Consistent with the known data for all MM-patients, a higher stage according to ISS or evidence of one of its single factors was associated with impaired survival in various studies [52,55,73].

In line with a suspected higher GvM effect, occurrence of mild or moderate chronic GvHD led to a survival benefit [52,62,67,72,74]. On the contrary, the outcome was worse in patients developing severe acute GvHD, likely due to prolonged immunosuppression and increased TRM [52,71,73].

Expectedly, younger patient age [73,75,78,80,82], a good performance status [43] and participation in clinical trials [44] were found to be associated with a better outcome.

Several analyses proved a survival benefit after allo-SCT in case of a cytogenetic standard risk (SR) [17,29,48,52,77], or rather no disadvantage for HR aberrations [45,81,82], indicating that the dismal prognosis of HR cytogenetics may be overcome by allo-SCT and providing support for the use of allo-SCT in eligible HR patients. In contrast, a pooled analysis of 61 trials showed no difference in survival of SR patients after auto/allo- as compared to a tandem auto-SCT [51].

## 5. Consolidation and Relapse Therapy after Transplantation

Due to immunological synergies in the post-transplant setting, the combination of allo-SCT with novel agents, such as PI, IMID, monoclonal or bispecific antibodies, antibody drug conjugates, CAR-T cells and/or DLI in relapsed patients or as consolidation therapy seems very promising.

Similar to the prognostic factors identified in the pre-transplant setting, an improved post-relapse survival after allo-SCT was demonstrated in case of cytogenetic SR, a long interval between allo-SCT and relapse, the absence of acute GvHD and the occurrence of (milder) chronic GvHD [71,74]. A pooled analysis of four prospective trials conducted in the first-line setting demonstrated an enhanced post-relapse survival after auto/allo-SCT as compared to a tandem auto-SCT [35,44], indicating a sustained immunological effect. However, this difference was not observed in a fifth prospective first-line trial [26], and not in case of HR cytogenetics [29]. The addition of novel agents, in particular IMID and PI, in the induction therapy and after allo-SCT was identified as a beneficial prognostic factor in several retrospective analyses [44,74,79,80], and response to PI and IMID was similar no matter if these substances were applied in pre- or post-transplant settings [75].

A post-transplant consolidation with DLI can also boost the donor immune system, but may induce an increased GvHD-risk [75]. In patients relapsing after allo-SCT DLI alone [83–85] or in combination with IMID and PI [36,86] led to a sustained anti-myeloma effect.

IMID-induced stimulation of alloreactive lymphocytes may improve response rates both applied for maintenance or post-transplant relapse, but may also augment GvHD. Indeed, in a trial concerning Lenalidomide-maintenance, acute GvHD led to study discontinuation in almost 40% of the patients [24,87–93]. The post-transplant application of PI as maintenance or relapse therapy, mostly Bortezomib, but also Ixazomib, is promising due to their intrinsic anti-myeloma effect and a possible suppression of GvHD without offsetting the GvM effect [32,61,82,86,90,91]. Thus, the combination of Lenalidomide and Bortezomib has also been discussed to sustain anti-myeloma effects and avoid GvHD [94].

Preliminary data have also shown promising responses after application of the CD38-antibody Daratumumab in MM-patients relapsed after allo-SCT with acceptable toxicity [95,96].

So far, no data has been published about the use of CAR-T cells [97], immunoconjugates or bispecific antibodies directed against MM-cells in the post-transplant setting. However, the possible synergistic immune effect of this therapy sequence and its tolerability should be clarified, and also whether allo-SCT in the era of further improved CAR-T cells may even be more rarely applied in the future.

## 6. Recommendations and Future Perspectives

In the past years, the therapeutic approaches for patients diagnosed with MM and their prognosis have decisively changed with the development of highly efficient new anti-myeloma drugs, such as PI, IMID, monoclonal antibodies and CAR-T cells, thus the role of allo-SCT has to be reevaluated in this context. Due to the GvM effect, it may allow long-term survival and probably even cure, but is associated with a considerable toxicity and has to be carefully evaluated in suitable young and fit patients with risk factors in the initial course of therapy. The combination of auto- and allo-SCT with RIC-regimens has shown survival benefits for HR patients in the first-line setting, albeit current data are inconsistent, and it is not routinely conducted in clinical practice outside clinical trials (Figure 1). Salvage allo-SCT is recommended, preferentially within clinical trials, for patients with early relapse after first-line therapy including auto-SCT and in HR constellations according to cytogenetics and stage (Figure 1).

Current T-cell based immunotherapeutic approaches lead to highly promising response rates, but do obviously not induce long-lasting disease control [98]. Thus, allo-SCT may remain a relevant therapeutic option in MM that should be discussed in certain carefully selected cases.

Future prospective trials are warranted especially to define the role of salvage allo-SCT in patients with RRMM and to examine risk-adapted protocols including allo-SCT with RIC-regimens in combination with new immunotherapeutic agents that can lead to a sufficient cytoreduction before allo-SCT and enhance the GvM effect after transplantation, and thus may allow a long-term disease control, preservation of patients' quality of life and prolonged survival. Due to the heterogeneity of the disease, various patient- and disease-specific factors have to be considered in the study design like R-ISS-criteria, especially certain genetic markers, radiomics and response evaluation including MRD-assessment [99], to identify those HR patients that may benefit most from allo-SCT. In addition to this individual risk stratification, optimization of conditioning protocols and GvHD-prophylaxis seems essential to further reduce therapy-related toxicity.

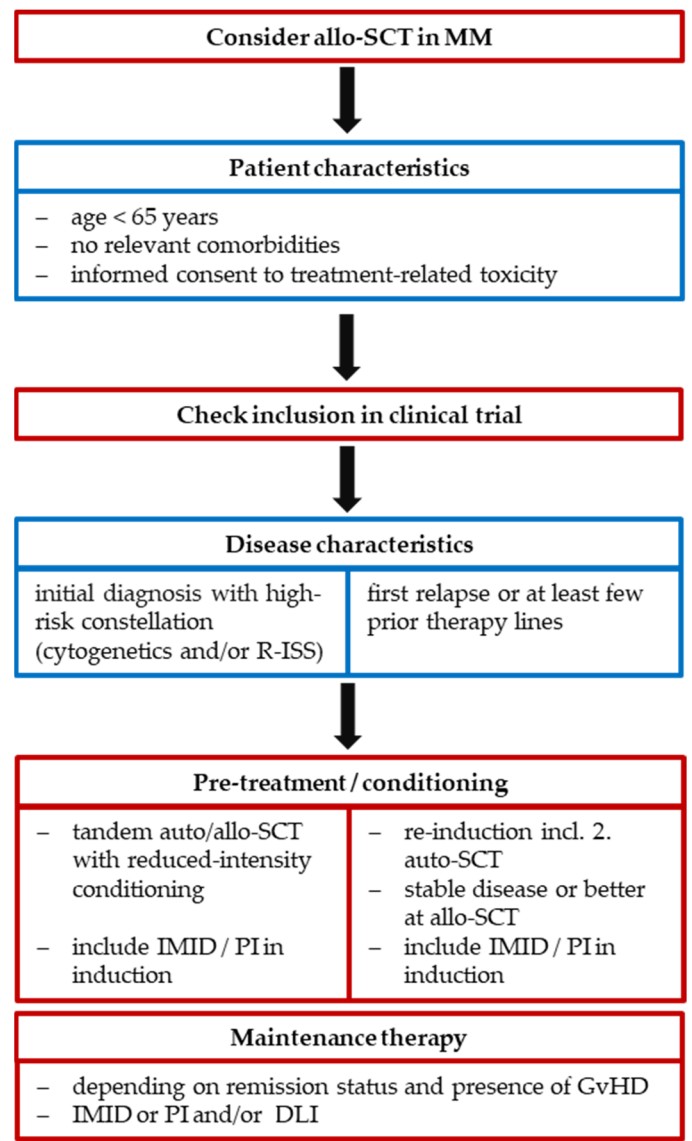

**Figure 1.** Consideration criteria for allo-SCT in MM. Abbreviations: R-ISS = revised international staging system; auto/allo-SCT = autologous/allogeneic hematopoietic stem cell transplantation; MM = multiple myeloma; DLI = donor lymphocyte infusions; IMID = immunomodulatory drugs; PI = proteasome inhibitors; GvHD = graft-versus-host disease.

**Author Contributions:** Conceptualization: C.G. and R.W.; Literature research: C.G.; Draft preparation: C.G. and R.W.; Review and editing: C.G., M.E., J.F., and R.W. All authors have read and agreed to the published version of the manuscript.

**Funding:** This research received no external funding.

**Institutional Review Board Statement:** Not applicable.

**Informed Consent Statement:** Not applicable.

**Conflicts of Interest:** The authors have no conflict of interest to declare.

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
