# Peer review of "Allogeneic Stem Cell Transplantation in Multiple Myeloma"

_cancers, doi:10.3390/cancers14010055_

Round 1
Reviewer 1 Report
This is a well-written article that comprehensively analyzes the flow of treatment development for allo-SCT treatment for MM.Thank you for your nice contribution.
Reviewer 2 Report
Greil and her colleagues reviewed the state of the art of allogeneic stem cell transplantation for patients with multiple myeloma. The review is comprehensive. My understanding is that the authors would recommend allo HSCT for young and fit patients with high risk myeloma as first line treatment, and for young and fit patients with standard risk myeloma as second line of treatment.
Minor comment :The figure at the end of the paper tries to summarize this conclusion. It remains unclear wether the proposition would be in the context of a clinical trial. I would suggest to present the proposition as a decision tree.
In addition, I would also interested wether preemptive post-transplant has been used or will be added
Reviewer 3 Report
Outstanding and comprehensive review of the role of allo transplant in multiple myeloma from authors experience. This work can provide a useful reference for discussion. The field of myeloma treatment has been rapidly evolving and new effective agents continue to enter the pipeline. As the authors note, however, this means that the role of allo transplant in myeloma is a moving target, difficult to achieve consensus on. I agree with the authors that allo transplant could have a role in regimen with the goal of cure.
One issue that could be better addressed is the trade off between quality of life with GVHD and OS. How many surviving allo patients have cGVH in your experience?
My primary concern is the lack of detail from these experts on their recommendations. Figure 1 as it stands is not helpful. If greater detail were added however this could be a powerful addition to the manuscript.
- Is the left side of Figure 1 the way you practice? This should be made clear.
- What defines young, how are fitness and comorbidities measured?
- Consider defining the age of "young," i.e. who would benefit from allo transplant, by analyzing OS benefits. 55 years old or less would be my guess, but how to rationally choose this cut off?
- Take the question mark off of "clinical trial" and make the right side of Figure 1 clearer with regards to eligibility criteria and schema you suggest for a clinical trial of allo in myeloma
- Personally, I would not allo transplant ANY myeloma patient up front
- I would be enthusiastic for a trial for patients under 55 or so who relapse after aggressive immunotherapy based induction chemotherapy
Reviewer 4 Report
Christine Greil et al uncovered Allogeneic stem cell transplantation to potentially enable long-term survival improvement in MM.
Point to be considered:
1) The rationale of why the authors came up with this review.
2) What is the information that is not exactly available that motivated the authors to come up with this information. What are the current caveats and how do the authors highlight the current research in answering them? If not they need to address in future directions.
3)In young high-risk RRMM, if we consider that the life expectancy would be 5 to 10 years, the patient could be motivated to do Allo (for a 36 y. pt would be a maximum life expectancy of 46), in this case, data from BMTCTN0102 data with long follow up, the PFS is significantly better with AUTO/ALLO over a tandem auto. What about OS? Can the author comment on this?
4)F. Patriarca et al uncovered having a donor vs. non donor improved OS because these patients would undergo Allo. But should we propose the standard allo we have done? In other words, we push the cell, we pull the drug, and we pray that all works out? Probably we should consider allo- with investigational strategies, i.e. intense conditioning regimen (Bus-Mel-Flu) followed by a cd34 selected allograft to prevent GVHD: this can be a platform to add on a further immunotherapy. Morever, reduced intensity conditioning with Flu and TBI and anti-CD45 monoclonal antibody radio-immunoconjugates can represent an option. In the era of CAR-T cells BiTEs, ADC, how would the authors comment on these bullets?
5)The underlying message here is that more precision and individualized approaches need to be tested in well designed clinical trials – a challenge, but I would be interested in their perspective of how this might be done (i.e. refer to PMID: 31323969, figure 5 and discuss).
6) The authors need to highlight what new information the review is providing to enhance the research in progress.
Round 2
Reviewer 3 Report
Well done
Reviewer 4 Report
The authors have clarified several of the questions I raised in my previous review. Most of the major problems have been addressed by this revision.